# HIV knowledge and reported stigma among South Florida health fair participants

**Sophia J. Peifer**[1], **David Mitchell Moore**[1], **Raphael Lee**[1], **Daniel J. Feaster**[2], **Candice A. Sternberg**[1,3]*

**1** University of Miami Miller School of Medicine, Miami, Florida, United States of America, **2** Department of Public Health Sciences, University of Miami Clinical Research Building, Miami, Florida, United States of America, **3** Department of Medicine, University of Miami Miller School of Medicine, Miami, Florida, United States of America

* c.aurelus@miami.edu

## Abstract

### Background

Florida had the third greatest number of new HIV diagnoses in the United States in 2020. This cross-sectional study aimed to assess HIV education and perceptions among diverse populations in South Florida to enhance public health community outreach efforts. Specifically, it investigated how HIV knowledge and perceptions vary based on race, primary language, and country of origin.

### Materials and methods

Cross-sectional surveys were administered at five South Florida health fair locations to evaluate understanding of HIV transmission, strategies for prevention and treatment, and stigma among those who accepted and declined free HIV testing. We analyzed survey data using chi-square tests with an alpha level of 0.05.

### Results

Of the 173 respondents, 149 underwent HIV testing, while 24 declined. Out of all respondents, 20.8% identified as Black (n = 36), 29.5% White non-Hispanic (n = 51), and 43.9% White Hispanic (n = 76). Over half of all respondents were foreign born (59%). Most participants knew HIV can be spread by injection drug usage (98.8%) and unprotected sex (97.7%). Incorrect answers included that HIV could be spread by mosquito bites (27.2%), kissing a person living with HIV (26.6%), and sharing a drink with a person living with HIV (19.7%). Transmission knowledge was significantly associated with race ($\chi^2$(2, N = 163) = 8.78, p = .012), with 26.3% Black (n = 10), 18.7% White Hispanic (n = 14), and 4.0% of White non-Hispanic participants (n = 2) reporting inadequate transmission knowledge. Familiarity with PrEP and/or PEP was also associated with race ($\chi^2$(2, N = 163) = 13.27, p = .001), as White Hispanic participants

**Data availability statement:** The data file with de-identified survey responses is available at the link: http://datadryad.org/share/8Evolk-NA6a7g6L5s5UcxP8MR9IHfII3O7UWuyXGF-6GM.

**Funding:** Dr. Sternberg is funded by the National Center for Advancing Translational Sciences, award number KL2TR002737.

**Competing interests:** No competing interests to disclose.

reported the highest lack of familiarity (84.2%), and Spanish-speaking participants reported half the PrEP/PEP familiarity as their English-speaking counterparts (p < 0.0001).

## Conclusion

Transmission knowledge was significantly low among Black and White Hispanic participants, while PrEP/PEP knowledge was uniquely low among White Hispanic and Spanish-speaking participants, reinforcing the need for improved education among these populations.

## Introduction

The HIV epidemic currently affects more than 38 million people worldwide and over 1 million people in the United States [1,2]. HIV disease burden is significantly greater in the South compared to other geographical regions [3]. Furthermore, individuals in the Southern United States are less likely to know their HIV status or receive timely treatment. In 2020, Florida had the third greatest number of new HIV diagnoses in the United States with 3,258 new cases [1]. HIV diagnoses disproportionately affect Black Florida residents, as 43.2% of people living with HIV (PLWH) in Florida are Black, 26.8% are Hispanic, and 26.7% are White [4].

Even though HIV continues to affect many individuals, there are effective methods of prevention and treatment. Pre-exposure prophylaxis (PrEP) is a medication regimen recommended to prevent HIV among individuals who may be at increased risk. Similarly, post-exposure prophylaxis is a treatment taken after a known or suspected HIV exposure. Over a decade ago, the United States Food and Drug association approved PrEP with oral tenofovir and emtricitabine as an effective method to prevent HIV transmission [5]. Likewise, post-exposure prophylaxis (PEP) with tenofovir, emtricitabine, and either raltegravir or dolutegravir is approved to prevent HIV after a potential exposure, if taken within 72 hours [6]. Given advancements in HIV treatment, patients who promptly start triple anti-retroviral therapy (ART) have the same lifespan as individuals not living with HIV [7]. There are numerous barriers that prevent individuals from utilizing PrEP, PEP, or ART including cost, education, stigma, distance to facilities, and pharmacy availability [8–11]. Given these access barriers, Black PLWH are less likely than White PLWH to achieve viral suppression in Florida [12].

Despite advancements in HIV treatment and prevention, misconceptions continue to prevail in the general population. Many of these misconceptions stem from misinformation spread during the 1980s AIDS epidemic when the virus was poorly understood [13]. In the United States, people continue to hold misconceptions that HIV can be transmitted through sneezing, sharing a drink, using public toilets, or mosquito bites. [14]. Focus groups revealed that older adults continue to worry about HIV transmission via casual contact [15]. These false beliefs contribute to HIV-related stigma and decrease utilization of community screening, care establishment, and treatment adherence [16,17].

Not only do transmission misconceptions limit patient utilization of HIV testing, but stigma also contributes to the limited use of prevention services [18]. Stigma can be defined as personal, when people harbor negative beliefs or attitudes towards PLWH, or perceived, when people perceive a societal prejudice against PLWH but do not personally harbor negative opinions [19]. In general, stigma is reinforced by societal structures and dependent on social interactions, witnessed acts of marginalization, misinformation, and discriminatory laws or policies [20]. Global studies have demonstrated that HIV-related stigma persists due to the association of HIV with marginalized behaviors like sex work, intravenous drug use, and anal sex [21,22]. Studies show that increased perceived and personal stigma are directly correlated with delayed care utilization and decreased ART usage among PLWH [23,24].

To combat HIV-related stigma, there have been numerous culturally tailored interventions targeted at various groups. Our prior research in South Florida aimed to better understand educational gaps about PrEP usage among Black non-Hispanic individuals to tailor culturally competent interventions [25]. Likewise, we performed interviews among Black non-Hispanic individuals in South Florida to better understand barriers to telehealth use for HIV care among this population and assess the effectiveness of these services [26]. These and other studies reiterate the need for cultural competence when designing interventions to increase HIV education and access among Black individuals and immigrants, which has been further supported by literature reviews aimed at better characterizing HIV testing among immigrants [27].

Limited studies have investigated the knowledge of HIV among Hispanic residents of South Florida. A past cross-sectional study indicates that 88.5% of Hispanic women in South Florida did not perceive themselves to be at risk for acquiring HIV, despite having a three times higher risk than that of White women [28]. Likewise, few studies have investigated knowledge among Black South Florida residents. Previous data shows that few Black South Florida residents understood the difference between HIV and AIDS or knew that medications could extend the lives of persons living with HIV [29]. Overall, these studies suggest that education aimed to eliminate HIV-related misconceptions could be useful in these communities.

More recent research is needed to better understand HIV knowledge and perceived stigma among South Florida communities seeking testing. Most prior research has been aimed at characterizing knowledge and stigma among PLWH, rather than members of the general community. There are few studies that characterize HIV-related stigma, HIV knowledge, and willingness to get tested among the same respondents. This cross-sectional study aimed to assess HIV education and perceptions among diverse populations in South Florida to enhance public health community outreach efforts. Specifically, it investigated how HIV knowledge and perceptions vary based on race, primary language, and country of origin.

## Materials and methods

### Research approach and design

A survey instrument was developed consisting of 23 questions adapted from previously validated scales that measured HIV knowledge and stigma among the general public (S1 Appendix) [30,31]. Questions were adapted for relevance and those deemed more pertinent were selected to avoid respondent fatigue. This was a cross-sectional survey utilizing a quantitative approach administered from October 15th, 2022 – April 15th, 2023 to participants who provided verbal informed consent at five health fair locations across Miami-Dade and Monroe counties: Kendall, the Florida Keys (Marathon, Big Pine, and Key West), Liberty City, Allapattah, and Homestead. Verbal consent was witnessed by a trained medical student volunteer and documented by typing the participant's name in the survey form. All surveys were translated from English to Spanish and Haitian Creole for administration in the participant's native language in compliance with IRB standards (IRB#20220835), and then back translated to ensure accurate translation by a certified translator. Among those who received HIV testing, participants completed the survey while awaiting results. Participants were either given the option to read the survey independently or to have the volunteer read the survey to them. However, all participants were directed to

complete the four stigma questions privately, as to not bias results. After completing the survey, volunteers explained any incorrectly answered questions for participant education purposes.

## Population, sample size, and sampling

All health fair participants who met HIV screening criteria according to the United States Preventive Services Task Force (USPSTF) guidelines, regardless of if they opted for HIV testing, were asked if they would like to participate in the survey. For those who opted out of HIV testing for any reason, but consented to answer a survey, a slightly modified survey was administered (S2 Appendix). We included health fair participants over the age of 18 capable of providing informed consent. No participants who completed the survey were excluded from analysis. Statistical power was assessed using Pass2020 with an alpha rate of.05 [32]. We examined power for sample sizes from 150 to 200, using Cohen's was the effect-size measure with a sample of 173. There is 80% power to uncover a chi-square with 1 degree of freedom with a w of.213. Cohen (1988) characterized a w = .10 as a small and a w = .30 as a medium effect, so this study has 80% power to uncover a small to medium effect [33].

## Measures

**Survey composition.** Survey questions are grouped into five categories: demographics, HIV transmission knowledge, HIV prevention and treatment knowledge, personal stigma, and perceived stigma. HIV transmission knowledge and HIV stigma are the main constructs measured using previously validated instruments. One third of the HIV transmission knowledge questions are adapted from the Marcelin et al. study and two thirds of the transmission knowledge questions are derived from the Herek et al. study [14,31,]. All of the personal/perceived stigma questions are adapted from the STRIVE stigma questionnaire [34]. The Herek et al. study, has a high internal consistency (α = .77−.79) but there are no reported internal consistencies reported for the Marcelin et al. or STRIVE stigma questionnaires [14,31,34].

Participants are categorized as White non-Hispanic, White Hispanic, or Black. Only two participants identified as Black Hispanic and are classified as mixed race. All participants described as Black are non-Hispanic. If participants indicated that they speak two or more languages, they are analyzed separately as bilingual. Initially, gender, age, and educational demographic information was not collected; however, this data was incorporated after the first two health fairs. The transmission knowledge section gauged participants understanding of how HIV can be transmitted with six true/ false questions adapted from prior surveys: 1) HIV can be spread by needle-sharing, 2) HIV can be spread by unprotected vaginal sex, 3) HIV can be spread by unprotected anal sex, 4) HIV can be spread by kissing, 5) HIV can be spread by drinking from the same glass, and 6) HIV can be spread by mosquitoes. The next survey questions gauge participants' familiarity with medical prevention/ therapy strategies, namely PrEP, PEP, and anti-retroviral therapy. Participants indicated yes or no to "I have heard of post-exposure prophylaxis (PEP)" and "I have heard of pre-exposure prophylaxis (PrEP)".The final four questions are aimed at gauging participants personal stigma and perceived stigma of others towards persons living with HIV, a surrogate for assessing potential stigma in certain communities. Participants demonstrate personal stigma if they screen positive on the first two questions indicating that they had less respect for people living with HIV or believed that people living with HIV deserved their situation. Participants demonstrate perceived stigma if they screened positive on the last two questions indicating that they believe other people talk badly about people living with HIV or that they would be worried if their friends and/or family found out that they had HIV.

**Survey scoring.** Transmission knowledge scores are classified as adequate if participants missed no more than one question out of the six total. This cut off was determined based on pilot data from the initial health fair survey responses (n = 12), which demonstrated a mean of 80% correct. Participants are considered familiar with PrEP/PEP if they noted familiarity with either medication on the questionnaire. For the four stigma questions, participants demonstrated personal stigma if they screened positive on one of the first two questions or perceived stigma if they screened positive on at least one of the last two questions.

**Statistical analysis.** Qualtrics was used to collect participant responses, and then the raw data was exported to Microsoft Excel. Response rates to each survey question were then aggregated in Excel according to the methods above and imported into GraphPad prism for statistical analysis. All data was analyzed using chi-square analysis with 1 degree of freedom at an alpha level of 0.05. We calculated the internal consistency of our study by performing Cronbach's alpha analysis on PEP/PrEP treatment knowledge questions and calculated alpha coefficients of 0.75.

## Results

Out of 173 respondents, 149 received HIV testing while 24 declined HIV testing. Surveyed participants ranged between the age of 18 to over 75 years old, with the largest groups of participants aged 50–59 (24.8%) and 60–75 (19.1%) (Table 1). 36 respondents were Black (20.8%), 51 were White non-Hispanic (29.5%), and 76 White Hispanic (43.9%). 59% of respondents were foreign born, with the most common countries being Cuba (n = 11), Mexico (n = 11), Venezuela (n = 10), and Haiti (n = 9). With respect to primary language spoken at home, 85 spoke primarily English (49.1%), 56 spoke primarily Spanish (34.78%), 11 spoke both English and Spanish (6.4%), and 9 spoke primarily Haitian Creole (5.2%) (Table 1). Among those with complete demographics, most participants were female (46.8%) and attended at least some college (50.3%).

Consistent across all demographics, over 97% of participants were familiar with the primary modes of HIV transmission: injection drug usage and unprotected sex, both vaginal and anal (Fig 1A). However, a significant percentage of participants additionally believed that HIV could be spread by mosquito bites (27.2%), kissing a person living with HIV (26.6%), and sharing a drink with a person living with HIV (19.7%). More than 75% of participants indicated that they had never heard of PrEP or PEP (Fig 1B). Nearly half (48%) were unaware that people living with HIV can live long and healthy lives on antiviral therapy.

Over half of participants reported perceived community stigma while only a small percentage of participants reported personal stigma.

Transmission knowledge was significantly different by race ($\chi^2$(2, N = 163) = 8.78, p = .012) (Table 2), with 10 Black (26.3%), 14 White Hispanic (18.7%), and 2 White non-Hispanic participants (4.0%) reporting inadequate transmission knowledge (Fig 2A). Similarly, familiarity with PEP and/or PrEP was also associated with race ($\chi^2$(2, N = 163)= 13.27, p = .001), with a significant 84.2% of White Hispanic participants (n = 64) reporting lack of familiarity, as opposed to 33 White non-Hispanic participants (64.71%) and 19 of Black (52.8%) participants. Among White non-Hispanic and Black participants, there was no statistical significance (Fig 2B).

While there was no difference in HIV transmission knowledge between English and Spanish speaking participants, PrEP/PEP familiarity was significantly greater among English-speaking participants compared to Spanish-speaking participants ($\chi^2$(1, N = 137) = 9.20, p = .002) (Fig 3). Even when participants spoke both English and Spanish, they demonstrated an almost identical level of lack of familiarity with PEP and/or PrEP (85%), but the cohort was small (n = 14). Interestingly, participants born in a Spanish-speaking country, including participants who spoke both English and Spanish fluently, had a lower knowledge of PrEP and/or PEP than those born in the United States (US) ($\chi^2$(1, N = 137) = 11.09, p < .001). There was no significant difference in the rates of college educated individuals broken down by race ($\chi^2$(2, N = 116) = 3.85, p = .146), language ($\chi^2$(1, N = 97) =.01, p = .938), or country of origin ($\chi^2$(1, N = 98) =.68, p = .408). Perception of risk, as determined through the question "I believe I am at risk for HIV", showed no statistical association with participant transmission or PrEP and/or PEP knowledge.

Few participants indicated personal stigma regarding those living with HIV (13.9%) (Fig 1C). Among this small sample (n = 24), there was about a two-fold increase in personal stigma rates between English and Spanish speaking participants (10.6% vs 21.15% respectively), but this did not reach statistical significance. Perceived and personal stigma had a low Cronbach's alpha coefficient value and thus we drew limited conclusions about their association with race, country of origin, and primary language. Furthermore, personal stigma rates were not significantly different between participants with

**Table 1. Sample Population Demographics.**

| Demographic | Number (n = 173) | Percentage |
|---|---|---|
| **Age Range** | | |
| Under 30 | 26 | 15.02% |
| 30–49 | 60 | 34.68% |
| 50–59 | 45 | 26.01% |
| Over 59 | 42 | 24.28% |
| **Gender** | | |
| Male | 43 | 24.86% |
| Female | 81 | 46.82% |
| Did not answer | 49 | 28.32% |
| **Race** | | |
| Black | 36 | 21.05% |
| Non-Hispanic White | 51 | 29.82% |
| Hispanic/Latino | 76 | 44.44% |
| Asian | 5 | 3.36% |
| Two or more Races | 3 | 1.34% |
| Unknown | 2 | 1.16% |
| **Region of Origin** | | |
| US | 71 | 41.04% |
| Central America | 23 | 13.29% |
| South America | 31 | 17.92% |
| Caribbean | 29 | 16.76% |
| Europe | 13 | 8.61% |
| Other | 6 | 3.97% |
| **Highest Level of Education** | | |
| Graduate School | 11 | 6.36% |
| College | 53 | 30.64% |
| Some College | 23 | 13.29% |
| High School | 28 | 16.18% |
| Some High School | 5 | 2.89% |
| Did not answer | 53 | 30.64% |
| **Primary Language** | | |
| English Only | 85 | 52.80% |
| Spanish | 56 | 34.78% |
| Both English and Spanish | 11 | 6.83% |
| Creole Only | 5 | 3.11% |
| Both English and Creole | 4 | 2.48% |
| Other | 12 | 6.94% |

adequate and inadequate HIV transmission knowledge. A greater number of participants reported perceived stigma by their community (59.0%) and family members (31.8%) (Fig 1C).

## Discussion

This cross-sectional study aimed to assess HIV education and perceptions among diverse populations in South Florida to enhance public health community outreach efforts. Specifically, it investigated how HIV knowledge and perceptions vary

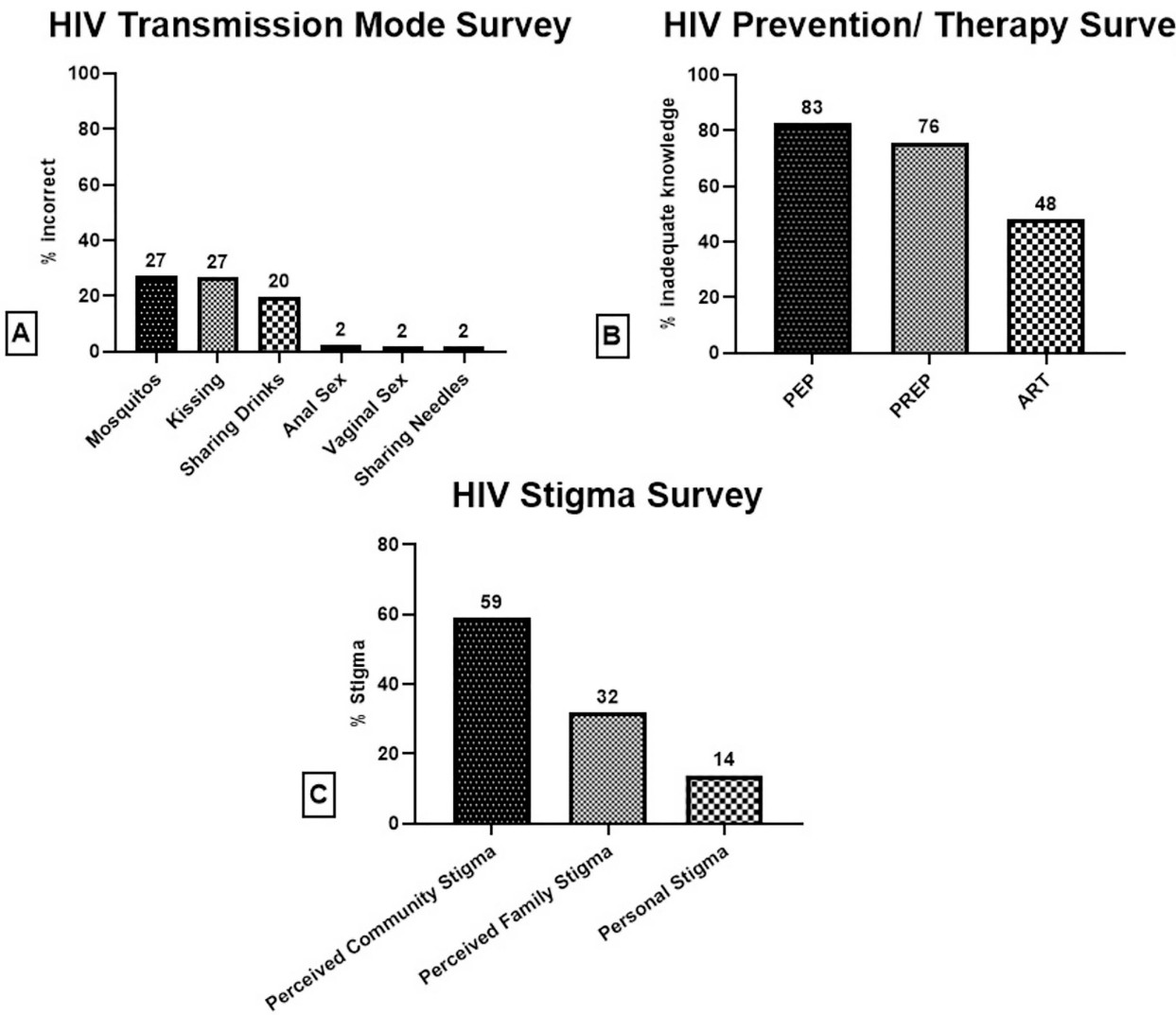

**Fig 1. Percentage Breakdown of Answers to Survey Questions Assessing Awareness of HIV Transmission Modes (A) Prevention/ Therapeutic Modalities (B) and Stigma (C).** Most common misconceptions were that HIV can be spread via mosquitoes and kissing. A majority of surveyed participants indicated they had no knowledge of PrEP or PEP, and close to half were unaware that PLWH could live long lives on antiretroviral therapy.

based on race, primary language, and country of origin. Through identification of these knowledge gaps, we aimed to improve education and outreach efforts to increase future utilization of HIV testing and prevention services among various demographic groups. The results highlighted a dire need for improved community-wide education on HIV transmission, prevention, and treatment. While almost all participants understood that HIV can be spread through sex and injection drug use, many held additional misconceptions about how it is spread, and these misconceptions were more commonly held by Black and White Hispanic participants, compared to White non-Hispanic participants, reaffirming knowledge gaps demonstrated in prior studies [28,29]. PrEP/PEP knowledge was more limited among White Hispanic participants, and our results demonstrate a clear need for improved educational efforts targeted at these groups which have lower rates of PrEP/PEP utilization [35].

Overall, HIV education should be tailored to combat common misconceptions, rather than reiterate information that is commonly understood by the general population. Almost all participants in our study understood that HIV can be spread

**Table 2. Chi Square Results.**

| | Percentage | Number | Chi-Square | P-Value | Significant |
|---|---|---|---|---|---|
| **Adequate HIV Transmission Knowledge vs. Race** | | 163 | 8.784 | 0.0124 | Yes |
| Black | 73.68 | | | | |
| White Non-Hispanic | 96 | | | | |
| White Hispanic | 81.33 | | | | |
| **Adequate HIV Transmission Knowledge vs. Primary Spoken Language** | | 137 | 0.0096 | 0.92 | No |
| English | 77.65 | | | | |
| Spanish | 76.92 | | | | |
| **PrEP/PEP Familiarity vs. Race** | | 163 | 13.27 | 0.0013 | Yes |
| Black | 47.22 | | | | |
| White Non-Hispanic | 35.29 | | | | |
| White Hispanic | 15.79 | | | | |
| **PrEP/PEP Familiarity vs. Primary Spoken Language** | | 137 | 9.196 | 0.0024 | Yes |
| English | 40 | | | | |
| Spanish | 15.38 | | | | |
| **PrEP/PEP Familiarity vs. Country of Origin** | | 137 | 11.09 | 0.0009 | Yes |
| USA | 40.85 | | | | |
| Spanish Speaking Country | 15.15 | | | | |
| **HIV Transmission Knowledge vs. Stigma** | | 171 | 0.8782 | 0.3487 | No |
| Stigma | 12.12 | | | | |
| No Stigma | 87.88 | | | | |
| **Stigma vs. Race** | | 163 | 2.756 | 0.252 | No |
| Black | 18.42 | | | | |
| White Non-Hispanic | 7.84 | | | | |
| White Hispanic | 17.33 | | | | |
| **Stigma vs. Primary Spoken Language** | | 137 | 1.7 | 0.0892 | No |
| English | 10.59 | | | | |
| Spanish | 21.15 | | | | |

via unprotected sex or injection drug usage, and these points are commonly stressed in educational curriculums. In contrast, fewer educational curriculums emphasize that HIV cannot be spread through contact or mosquito bites, which are more common areas of misconception. A recent 2022 meta-analysis analyzed the effectiveness of sex education/HIV programs across the world and found that HIV prevention education alone, at least in its current form, may be ineffective in influencing people's behavior [36]. Similarly, previous data collected in South Florida showed that many Black residents were unaware that HIV is a manageable disease, and the researchers inferred that this limited knowledge may limit participants' utilization of prevention services [29]. It is essential that HIV educational programs are tailored in a culturally sensitive manner to effectively modify participants' knowledge and behavior.

Beside behavioral modifications, the use of PEP and PrEP has proven to be significantly effective in reducing risks of HIV transmission [37]. With prior studies indicating that one of the greatest barriers to utilization of PrEP and PEP is lack of awareness in certain communities, this presents a prime spot for public health intervention efforts [38,39]. The majority of studies assessing PrEP and PEP knowledge have surveyed PLWH and men who have sex with men (MSM). Studies among Hispanic MSM have revealed more limited knowledge about PrEP effectiveness and methods of access [40,41]. This knowledge gap about PrEP is likely greater among immigrant populations in the US, who commonly under-utilize PrEP services [42,32].

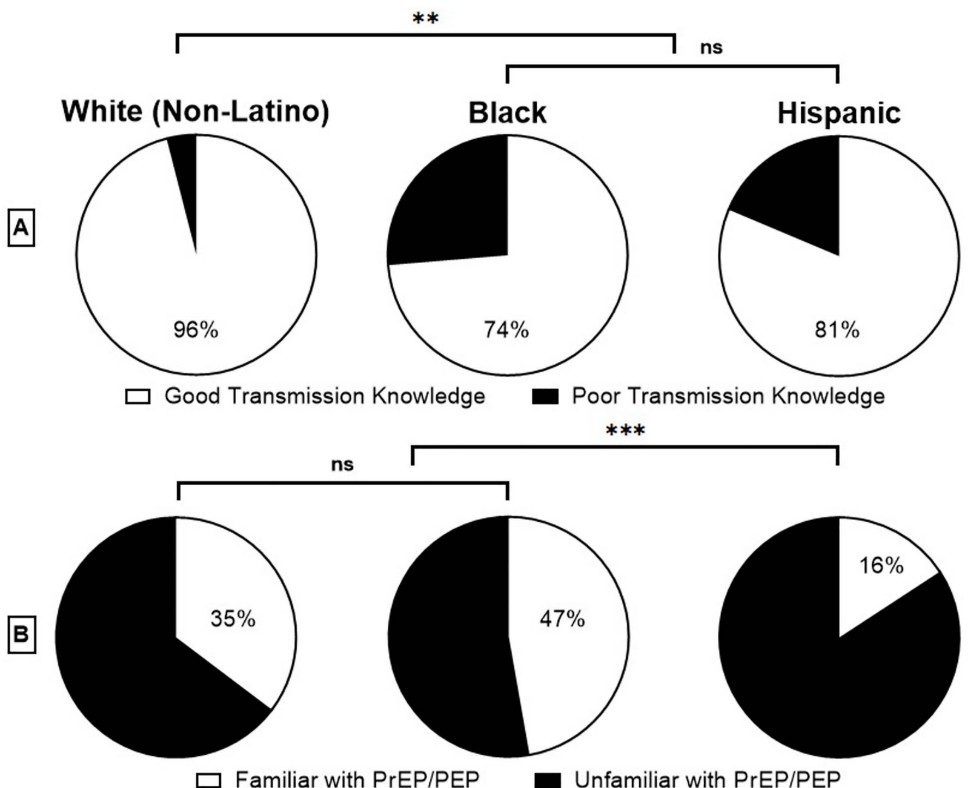

**Fig 2. Percent Transmission Knowledge (A) and Familiarity with PrEP/PEP (B) Based on Participant Racial Identity.** Hispanic participants exhibited significantly less knowledge about HIV transmission modes compared to their White counterparts **(A)**. Hispanic participants displayed less familiarity of PrEP and/or PEP compared to White and Black participants **(B)**.

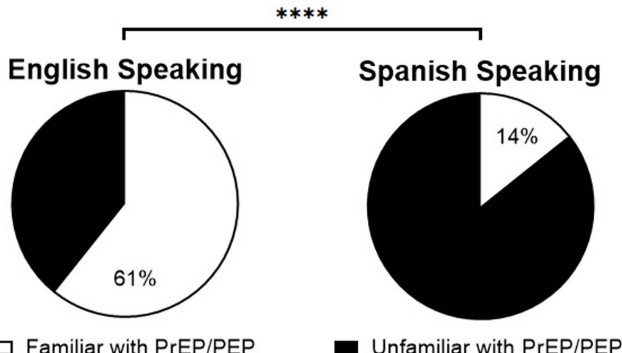

**Fig 3. PrEP/ PEP Knowledge based on Language Spoken.** Spanish speaking participants exhibited a significantly lower familiarity with PrEP and/or PEP than their English-speaking counterparts.

Our results indicated more limited PrEP and PEP knowledge among White Hispanic participants, but it is unclear whether the primary factor driving this difference is a foreign country of origin or Spanish as a primary language. PrEP/ PEP knowledge was greater among English-speaking participants than Spanish-speaking participants, but it was also greater among USA-born participants than participants born in Spanish-speaking countries. Supporting the hypothesis

that language is a driving factor is the finding that education levels did not significantly differ between groups, which might be expected when comparing participants born in the USA to those born outside. These results are consistent with a past study indicating that Spanish-speaking Latino men had reduced PrEP awareness compared to their English-speaking counterparts [40]. Furthermore, cultural distrust of medical care services and doctors within the Hispanic community likely contributes to these findings [43]. A qualitative study of Latino men who have sex with men found that there is skepticism about PrEP's effectiveness specifically among monolingual Spanish speakers [44]. Overall, we believe our data indicates that PrEP and PEP marketing companies should prioritize distributing translated and culturally sensitive materials to Hispanic communities in South Florida. In New York State, an enhanced PrEP Aware Week, utilizing social media messaging in both English and Spanish, increased PrEP prescription fills by 6–9%, indicating that culturally sensitive marketing can be incredibly effective [45].

Furthermore, our study demonstrates that personal stigma against PLWH may be greater among South Florida participants who speak Spanish. While few participants in our study indicated personal stigma, personal stigma was almost two-fold greater among Spanish-speaking participants, even though this did not reach statistical significance due to the small sample size, and further research is indicated. Past research has shown that perceived community stigma among Hispanic immigrants in the United States deters education, testing, and access to HIV prevention services among these communities [43,46–48]. In South Florida, past research indicated that HIV-related stigma was greater among respondents who indicated a country of origin other than the United States, or who interviewed in Haitian Creole or Spanish [49].

This study has some limitations, as all participants attending the health fair were invited to participate, most survey responses were from those who underwent HIV testing. Furthermore, the cross-sectional nature of the study only allowed us to sample health fair participants at one timepoint, limiting our sample size and ability to draw conclusions about the impact of HIV education. Moreover, we could not adequately assess bilingual or Haitian-Creole speakers given the low sample size. The survey required patients to self-report measures, which may have introduced some bias due to social desirability. This may have led to a selection bias, as it is possible that participants who refused both testing and the survey may have abstained due to personal stigma. Additionally, participants were likely inherently health conscious as they were sampled from preventative health fairs, which also promoted a selection bias. When evaluating survey responses, we utilized pilot data to determine the cut-off for sufficient transmission knowledge among the population which introduces variability. Another limitation was that at the first two health fairs, demographic information about gender, age, and educational level was not collected, limiting analysis of these variables. While we assessed interactions via bi-variate chi-square analysis, we did not run a multivariable logistic regression to assess confounders because numerous demographic variables including race, country of origin, and language are interrelated and cannot be considered independent variables.

## Conclusions

Misconceptions about HIV transmission and limited knowledge about PrEP and PEP usage continue to be widespread among White Hispanic and Black South Florida residents. While personal stigma against PLWH was minimally reported, many participants reported perceived stigma. Culturally sensitive interventions to combat HIV transmission misinformation and increase knowledge about PrEP and PEP can help to increase HIV prevention efforts in South Florida.

## Supporting information

**S1 Appendix. Survey Administered to Participants Undergoing HIV Testing.**
(DOCX)

**S2 Appendix. Survey Administered to Participants who Declined HIV Testing.**
(DOCX)

## Acknowledgments

We would like to thank Emmanuel Paul for his assistance with Haitian-Creole survey translations, Adrian Parra for his help with Spanish survey translations, Wendy Li and Nicholas Houser for their assistance with survey distributions, the University of Miami Writing center, the survey participants, and the University of Miami for their support of this project.

## Author contributions

**Conceptualization:** Sophia J. Peifer.

**Formal analysis:** David Mitchell Moore, Daniel J. Feaster.

**Investigation:** Sophia J. Peifer, David Mitchell Moore, Raphael Lee.

**Project administration:** Raphael Lee.

**Supervision:** Daniel J. Feaster, Candice Aurelus Sternberg.

**Validation:** Candice Aurelus Sternberg.

**Writing – original draft:** Sophia J. Peifer, David Mitchell Moore.

**Writing – review & editing:** Raphael Lee, Daniel J. Feaster, Candice Aurelus Sternberg.

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
