## [Decision Letter · Decision Letter 0]

11 Mar 2025

Dear Dr. Sternberg,

Thank you for submitting your manuscript to PLOS ONE. After careful consideration, we feel that it has merit but does not fully meet PLOS ONE’s publication criteria as it currently stands. Therefore, we invite you to submit a revised version of the manuscript that addresses the points raised during the review process.

Please submit your revised manuscript by Apr 25 2025 11:59PM. If you will need more time than this to complete your revisions, please reply to this message or contact the journal office at plosone@plos.org . A rebuttal letter that responds to each point raised by the academic editor and reviewer(s). You should upload this letter as a separate file labeled 'Response to Reviewers'.A marked-up copy of your manuscript that highlights changes made to the original version. You should upload this as a separate file labeled 'Revised Manuscript with Track Changes'.An unmarked version of your revised paper without tracked changes. You should upload this as a separate file labeled 'Manuscript'.

We look forward to receiving your revised manuscript.

Kind regards,

Zypher Jude G. Regencia, Ph.D.

Academic Editor

PLOS ONE

Journal Requirements:

3. In the online submission form, you indicated that your data is available only on request from a third party. Please note that your Data Availability Statement is currently missing [the name of the third party contact or institution / contact details for the third party, such as an email address or a link to where data requests can be made]. Please update your statement with the missing information.

“Dr. Sternberg is funded by the National Center for Advancing Translational Sciences, award number KL2TR002737.”

“Dr. Sternberg is funded by the National Center for Advancing Translational Sciences, award number KL2TR002737.”

Reviewers' comments:

Reviewer's Responses to Questions

**Comments to the Author**

1. Is the manuscript technically sound, and do the data support the conclusions?

Reviewer #1: Partly

Reviewer #2: Yes

Reviewer #3: Yes

2. Has the statistical analysis been performed appropriately and rigorously?

Reviewer #1: No

Reviewer #2: Yes

Reviewer #3: Yes

3. Have the authors made all data underlying the findings in their manuscript fully available?

Reviewer #1: No

Reviewer #2: Yes

Reviewer #3: Yes

4. Is the manuscript presented in an intelligible fashion and written in standard English?

Reviewer #1: Yes

Reviewer #2: Yes

Reviewer #3: Yes

Reviewer #1: Thank you for the opportunity to review this important research. Kindly address the following comments to ensure your paper meets PLOS One’s criteria for scientific suitability and publication:

General comments:

• The authors’ account for the unavailability of the data appears limited. While they mention that the data are available upon request, they do not provide sufficient details about the specific restrictions or exceptional circumstances that prevent public sharing.

• (Page 6, Line 119) “Materials and Methods:” All level 1-3 headings must be written in sentence case with only first word in capital letters.

Abstract:

• (Page 2, Line 30) The phrase “third greatest” should be written in words rather than using numerals (“3rd”) for formality.

• (Page 2, line 30-33) The background section of an abstract must incorporate aim to clarify specific purpose and direction of the study.

• (Page 2, line 34-37) Under Material and methods section, authors should briefly outline the approach, design, sampling method, and tool used for analyses, despite the word limit of the abstract section.

• (Page 2 Line 39) Change “For all respondents” since this is not a general statement but it a comparing from a group of participants, thus the use “Out of all respondents” would best fit this context.

• (Page 2 Line 40) Provide statistics from empirical data for this statement “Over half of all respondents were foreign born”.

• (Page 2, line 42) Authors must ensure consistent formatting by placing all percentages within parentheses.

• (Page 2, Line 44 & line 46-47) If these were results emanated from Chi-square analysis, author should include chi-square statistic value i.e., χ²(2, N = 173) = ….. p = .012.

Introduction:

• (Page 3, Line 59-60) clarify where “Deep South” is

• (Page 3, line 62) Write “3rd” in words such “Third” and include statistics that are indicative of a high prevalence of new HIV infections in this context.

• (Page 3, line 67) Define briefly and in simple terms “pre-exposure prophylaxis (PrEP)” and “post-exposure prophylaxis (PEP)”

• (Page 3, line 77-79) This sentence warrants citation “Many of these misconceptions can be traced back to the spread of misinformation during the AIDS epidemic in the 1980s, when little was known about the virus or its transmission.”

• (Page 4, line 79-83) Authors need to clarify the logic connected to how the misconceptions about HIV transmission could have the implications for community screening, care and ART adherence. For instance, how using public toilets, or mosquito bites could significantly discourage one from seeking community screening, pursing HIV care, and ART adherence. Does this mean, when people fear that casual contact could expose them to HIV, they may avoid testing and treatment? This may not always be the case, as some individuals might view these misconceptions as a health threat and still actively seek testing and treatment, motivated by the desire to safeguard their well-being despite their fears and false beliefs.

• (Page 4, Line 84-90) These sentences must be cited.

• (Page 4, Line 92-93) Authors might need to clarify the exact nature of the relationship since it is not clear whether this relationship is positive (both types of stigma lead to increased care and adherence) or negative (high levels of stigma lead to decreased care and adherence to ART). “Studies show that increased perceived and personal stigma are directly correlated with later care utilization and ART usage among PLWH. [19,20].

• (Page 5, line 99) Revise this phrase to best fit the context “These and other studies”.

• (Page 5-6, Line 115-118) The aim is very bread and vague “This cross-sectional study aims to gather information about HIV education and perception in South Florida to improve community outreach efforts, and better understand how HIV knowledge and perception may differ by race, primary language, and country of origin. Suggested revision: “This cross-sectional study aims to assess HIV education and perceptions among diverse populations in South Florida to enhance public health community outreach efforts. Specifically, it will investigate how HIV knowledge and perceptions vary based on race, primary language, and country of origin.”

Material and methods:

• (Page 6, line 120) Rephrase the following sentence for clarity “A 23-question survey was developed using questions adapted from previous validated surveys analyzing HIV knowledge and stigma among the public (S1 Appendix).” See suggested revision: “A survey instrument was developed consisting of 23 questions adapted from previously validated scales that measured HIV knowledge and stigma among the general public”.

• (Page 6, line 131-132) Back translation was performed to ensure the accuracy and cultural appropriateness of the translated materials? If yes, Authors should further elucidate this process and the No answer should be accompanied by explanation of how conceptual equivalence and content validity was ascertained on data collection instruments.

• (Page 6-7, line 120 -162) Authors should reorganize the manuscript under “Materials and methods” using level two subheadings for clarity and organization. Specifically, the following subheadings should be included:

1. Research Approach and Design: (Clearly define whether the study employed a quantitative approach using a cross-sectional design.)

2. Population, Sample Size, and Sampling: (Define the target population, explain how the sample size was determined using a specific formula with clear parameters, and state the sampling method used to select participants, justifying the choice of sampling method.)

3. Measures: (Describe and outline the items adapted from previously validated scales, report the internal consistency (Cronbach’s alpha coefficient) demonstrated by previous studies adopted the same scales, and include the internal consistency established by the current study.)

4. Analysis: (Detail the statistical methods used to analyze the data, including any software utilized, the specific tests conducted, and the criteria for significance.)

Results:

• (Page 8, line 164) Rephrase “Among 173 total respondents” to “Out of 173 respondents,…” and change “ and” to “while” in the following sentence for proper context “149 received HIV testing and 24 declined HIV testing.”

• (Page 8, line 166-171) all percentage values should be consistently placed within the parentheses for uniformity.

• (Page 8, line 165) Rephrase this part of the sentence for clarity “ranged in age from 18 to older than 75..” and change it to “…ranged between the age of 18 to over 75 years old,…”.

• (Page 8, line 176), In accordance with PLOS One guidelines, figures should be referenced in the text as (Fig. 1, Fig. 2, Fig. 3, etc.) instead of using the term “figure.”

• (Page 8-9, line 181-185) I strongly recommend authors to either discard this part or insert figure 1 as outlined and same comment applies to (Page 11, line 197 – 200) and (Page 12, line 213-215).

• (Page 11, line 201 – 202) Incorporate all values from the chi-square test for statistical difference /independence among these categorical variables. Further, this comment applies to all reported results without chi-square statistic.

• (Page 10, line 186) As a general guideline, the title for tables should be placed above the table.

• (Page 12, line 217) Rewrite “2-folds” as “two-folds” and this applies to (Page 14, Line 268)

• (Page 12, line 224) Rephrase “modified” to “influenced” throughout the manuscript.

Discussion:

• (Page 12, line 226-227) This study’s aim indicates a significant discrepancy between what is outlined in the last sentence of the introduction and the actual aim (to gather information while also now is to identify knowledge gaps). Authors should consider incorporating a consistent aim throughout to remind readers of the clear research direction and scope. Use the suggested statement of the aim.

• (Page 12, line 227-229) Revise this sentence in accordance with the revision made above.

• (Page 12, line 230) Authors should not introduce new concepts in the discussion for consistency, I thought initially it was “HIV transmission” not “HIV risk”.

• (Page 12-13, line 230-235) The discussion of the results should incorporate findings and outcomes from previous studies for comparison to provide context and highlight the value of the current research within the existing literature.

• (Page 13, line 237) Write “about three thirds” instead of “Close to 100%” since this is a discussion section.

• (Page 13, line 249) Rephrase this part of the sentence for clarity “usage of PEP and PrEP have shown” to “the use of PEP and PrEP has proven to be significantly effective in reducing risks of HIV transmission” and cite the source (s).

• (Page 14, line 256-259) Authors should clarify the connection between knowledge and language, as the study suggests that the lack of PEP and PrEP knowledge is somewhat influenced by language barriers. If the issue lies with language, it implies that the knowledge exists and is accessible but is hindered by challenges in decoding and interpreting it. Therefore, previous evidence should be cited to corroborate this assertion.

• (Page 14, line 261-263) This assertion should be supported by citations of scientific studies that have documented the connection between language and culture in the adoption of these preventive measures (PEP &PrEP) against HIV transmission.

• (Page 14, line 275) Authors should be careful not to use the term “screened positive” due to that stigma is typically measured using a validated scale rather than a screening tool. Various scales have been developed to assess different types of stigmas, such as HIV-related stigma, these scales include items that quantify the level of stigma an individual experiences or perceives, which makes it possible to measure the correlation between stigma and other variables. Thus, using a well-established stigma scale is not a limitation for measuring the correlation between stigma and other variables, as long as the scale is reliable and valid for the population being studied.

• Under limitations, authors should address the following aspects:

1. Design Used - The cross-sectional design adopted by the study

2. Self-Report measures - The use of self-report measures may introduce bias due to social desirability.

3. Sample size - The sample size may limit the generalizability of the findings, particularly if it is small.

Conclusion:

• (Page 15, line 286-287) “Stigma” is singular, so “was” should be used instead of “were.”

• (Page 15, line 287) “combat transmission” authors should specify that it “HIV transmission” for consistency.

References:

• (Page 17, reference 10) Authors should provide page number.

• (Page 17, reference 12) Authors should provide an issue number for this source.

• (Page 18, reference 21) Authors should provide an issue and page number for this source.

• (Page 19, reference 28) Authors should provide a page number for this source.

On the highest note: Additional recommendation:

Authors should provide a table that presents the results from the Chi-Square test of difference, clearly outlining the variables compared, the chi-square statistic (χ²), degrees of freedom (df), sample sizes, and p-values to highlight significant and non-significant differences.

Thank you.

Reviewer #2: There is no clear sample size calculation provided. Was the sample size of 173 participants determined based on power analysis? Since participants were recruited from health fairs, they may already be more health-conscious than the general population. A brief discussion on potential selection bias would strengthen the methodology section. The study considers race, language, and country of origin, but educational attainment could also significantly impact HIV knowledge. Were potential confounders controlled for? The study suggests that interventions should target Hispanic communities, but the data primarily focus on White Hispanic participants. Can these findings be generalized to all Hispanic groups, or are they specific to this subgroup?

Reviewer #3: Please provide the inclusion and exclusion criteria you used in recruiting survey respondents.

Among the respondents who consented to HIV screening, how many percent tested reactive, non-reactive? Perhaps it would be significant to assess their test results against their HIV transmission knowledge.

In Table 1, please include the key population in which the respondents qualify as (MSM, PWID, TGW etc).

**Do you want your identity to be public for this peer review?** For information about this choice, including consent withdrawal, please see our Privacy Policy

Reviewer #1: No

Reviewer #2: No

Reviewer #3: **Yes: ** Adrian Jonathan D. Velasco

---

## [Author Response · Author response to Decision Letter 1]

13 May 2025

May 8th, 2025

To: The Editor

Dear Dr. Regencia,

Thank you and the reviewers for your insightful review of our manuscript for publication in PLOS One. We have carefully considered this feedback and made changes to the manuscript. Below, we have addressed each comment and direct the reviewer to where they have been incorporated. Please update the funding statement as: “Dr. Sternberg is funded by the National Center for Advancing Translational Sciences, award number KL2TR002737.” Upon further consideration, we have shared de-identified data to promote additional research efforts and have uploaded the data to a repository (link below). http://datadryad.org/share/8EvolkNA6a7g6L5s5UcxP8MR9lHfll3O7UWuyXGF6GM

Sincerely,

Candice A. Sternberg, MD

Assistant Professor, Dept. of Medicine

University of Miami Miller School of Medicine

1120 NW 14th Street #858

Miami, Florida 33136

Phone: (305) 243- 2398

Email: c.aurelus@miami.edu

Editor:

https://journals.plos.org/plosone/s/file?id=wjVg/PLOSOne_formatting_sample_main_body.pdf
https://journals.plos.org/plosone/s/file?id=ba62/PLOSOne_formatting_sample_title_authors_affiliations.pdf

We have ensured that the manuscript complies with style requirements.

a. If there are ethical or legal restrictions on sharing a de-identified data set, please explain them in detail (e.g., data contain potentially identifying or sensitive patient information, data are owned by a third-party organization, etc.) and who has imposed them (e.g., a Research Ethics Committee or Institutional Review Board, etc.). Please also provide contact information for a data access committee, ethics committee, or other institutional body to which data requests may be sent.

There are no restrictions on sharing de-identified data.

b. If there are no restrictions, please upload the minimal anonymized data set necessary to replicate your study findings to a stable, public repository and provide us with the relevant URLs, DOIs, or accession numbers. Please see http://www.bmj.com/content/340/bmj.c181.long for guidelines on how to de-identify and prepare clinical data for publication. For a list of recommended repositories, please see https://journals.plos.org/plosone/s/recommended-repositories. You also have the option of uploading the data as Supporting Information files, but we would recommend depositing data directly to a data repository if possible.

The data has been uploaded to a repository with the link: http://datadryad.org/share/8EvolkNA6a7g6L5s5UcxP8MR9lHfll3O7UWuyXGF6GM

The data availability statement has been updated.

3. In the online submission form, you indicated that your data is available only on request from a third party. Please note that your Data Availability Statement is currently missing [the name of the third party contact or institution / contact details for the third party, such as an email address or a link to where data requests can be made]. Please update your statement with the missing information.

Upon reconsideration, we have decided to publicly share de-identified data. We have updated the online forms.

4. Please remove any funding-related text from the manuscript and let us know how you would like to update your Funding Statement. Currently, your Funding Statement reads as follows: “Dr. Sternberg is funded by the National Center for Advancing Translational Sciences, award number KL2TR002737.”Please include your amended statements within your cover letter; we will change the online submission form on your behalf.

Thank you. Funding information was removed from text and updated in the cover letter as requested.

Reviewer 1:

Thank you for your review. Your comments have greatly improved the manuscript.

General comments:

1. The authors’ account for the unavailability of the data appears limited. While they mention that the data are available upon request, they do not provide sufficient details about the specific restrictions or exceptional circumstances that prevent public sharing.

Upon reconsideration, we have decided to publicly share de-identified data to this link: http://datadryad.org/share/8EvolkNA6a7g6L5s5UcxP8MR9lHfll3O7UWuyXGF6GM

2. (Page 6, Line 119) “Materials and Methods:” All level 1-3 headings must be written in sentence case with only first word in capital letters.

Thank you. This has been corrected.

Abstract:

1. (Page 2, Line 30) The phrase “third greatest” should be written in words rather than using numerals (“3rd”) for formality.

This was clarified (page 2, line 30).

2. (Page 2, line 30-33) The background section of an abstract must incorporate aim to clarify specific purpose and direction of the study.

Thank you for this suggestion. We incorporated a sentence to describe the study aims that reads: “This cross-sectional study aimed to assess HIV education and perceptions among diverse populations in South Florida to enhance public health community outreach efforts. Specifically, it investigated how HIV knowledge and perceptions vary based on race, primary language, and country of origin.” (page 2, lines 31-34).

3. (Page 2, line 34-37) Under Material and methods section, authors should briefly outline the approach, design, sampling method, and tool used for analyses, despite the word limit of the abstract section.

Thank you for the suggestion. This section now reads: “Cross-sectional surveys were administered at five South Florida health fair locations to evaluate understanding of HIV transmission, strategies for prevention and treatment, and stigma among those who accepted and declined free HIV testing.” (Page 2, lines 35-37).

4. (Page 2 Line 39) Change “For all respondents” since this is not a general statement but it a comparing from a group of participants, thus the use “Out of all respondents” would best fit this context.

Thank you, this has been corrected (page 2, line 40).

5. (Page 2 Line 40) Provide statistics from empirical data for this statement “Over half of all respondents were foreign born”.

The sentence now reads: “Over half of all respondents were foreign born (59%)” (page 2, line 41).

6. (Page 2, line 42) Authors must ensure consistent formatting by placing all percentages within parentheses.

Thank you for the suggestion. The sentence has been rewritten: “Most participants knew HIV can be spread by injection drug usage (98.8%) and unprotected sex (97.7%) (page 2, line 41-43).

7. (Page 2, Line 44 & line 46-47) If these were results emanated from Chi-square analysis, author should include chi-square statistic value i.e., χ²(2, N = 173) = ….. p = .012.

Thank you for the suggestion. The chi-square statistic values are now reported in the abstract (page 2, lines 45, 48).

Introduction:

1. (Page 3, Line 59-60) clarify where “Deep South” is

The phrase was updated to “South” to better reflect the reference (page 4, line 57).

2. (Page 3, line 62) Write “3rd” in words such “Third” and include statistics that are indicative of a high prevalence of new HIV infections in this context.

This sentence now reads: “In 2020, Florida had the third greatest number of new HIV diagnoses in the United States with 3,258 new cases” (Page 4, lines 59-60).

3. (Page 3, line 67) Define briefly and in simple terms “pre-exposure prophylaxis (PrEP)” and “post-exposure prophylaxis (PEP)”

Information was added to explain PrEP and PEP in simpler terms: “Pre-exposure prophylaxis (PrEP) is a medication regimen recommended to prevent HIV among individuals who may be at increased risk of infection. Similarly, post-exposure prophylaxis is a treatment taken after a known or suspected HIV exposure” (page 4, lines 64-67).

4. (Page 3, line 77-79) This sentence warrants citation “Many of these misconceptions can be traced back to the spread of misinformation during the AIDS epidemic in the 1980s, when little was known about the virus or its transmission.”

Thank you for this suggestion. This sentence is now cited1 (pages 4-5, lines 77-78).

5. (Page 4, line 79-83) Authors need to clarify the logic connected to how the misconceptions about HIV transmission could have the implications for community screening, care and ART adherence. For instance, how using public toilets, or mosquito bites could significantly discourage one from seeking community screening, pursing HIV care, and ART adherence. Does this mean, when people fear that casual contact could expose them to HIV, they may avoid testing and treatment? This may not always be the case, as some individuals might view these misconceptions as a health threat and still actively seek testing and treatment, motivated by the desire to safeguard their well-being despite their fears and false beliefs.

Thank you for the suggestion. We edited the sentence and it now reads: “These false beliefs contribute to HIV-related stigma and decrease utilization of community screening, care establishment, and treatment adherence” (page 5, lines 81-83). Additionally, we added a source from the CDC which states that increased HIV stigma can discourage people from undergoing HIV testing.2

6. (Page 4, Line 84-90) These sentences must be cited.

Citations were added as follows: “Not only do transmission misconceptions limit patient utilization of HIV testing, but stigma also contributes to the limited use of preventative services.3 Stigma can be defined as personal, when people harbor negative beliefs or attitudes towards PLWH, or perceived, when people perceive a societal prejudice against PLWH but do not personally harbor negative opinions.4 In general, stigma is reinforced by societal structures and dependent on social interactions, witnessed acts of marginalization, misinformation, and discriminatory laws or policies5” (page 5, lines 84-90).

7. (Page 4, Line 92-93) Authors might need to clarify the exact nature of the relationship since it is not clear whether this relationship is positive (both types of stigma lead to increased care and adherence) or negative (high levels of stigma lead to decreased care and adherence to ART).

Thank you. This sentence was edited to state: “Studies show that increased perceived and personal stigma are directly correlated with delayed later care utilization and decreased ART usage among PLWH” (page 5, line 114-115).

8. (Page 5, line 99) Revise this phrase to best fit the context “These and other studies”.

Thank you, this was updated (page 5, lines 92-93).

9. (Page 5-6, Line 115-118) The aim is very bread and vague “This cross-sectional study aims to gather information about HIV education and perception in South Florida to improve community outreach efforts, and better understand how HIV knowledge and perception may differ by race, primary language, and country of origin. Suggested revision: “This cross-sectional study aims to assess HIV education and perceptions among diverse populations in South Florida to enhance public health community outreach efforts. Specifically, it will investigate how HIV knowledge and perceptions vary based on race, primary language, and country of origin.”

Thank you for the suggestion—this has been edited to incorporate the suggested wording (page 6, lines 115-118).

Material and methods:

1. (Page 6, line 120) Rephrase the following sentence for clarity “A 23-question survey was developed using questions adapted from previous validated surveys analyzing HIV knowledge and stigma among the public (S1 Appendix).” See suggested revision: “A survey instrument was developed consisting of 23 questions adapted from previously validated scales that measured HIV knowledge and stigma among the general public”.

Thank you. The suggested revision was incorporated (page 6, lines 121-122).

2. (Page 6, line 131-132) Back translation was performed to ensure the accuracy and cultural appropriateness of the translated materials? If yes, Authors should further elucidate this process and the No answer should be accompanied by explanation of how conceptual equivalence and content validity was ascertained on data collection instruments.

Thank you. More information was added on how the accuracy of the translated materials was confirmed: “All surveys were translated from English to Spanish and Haitian Creole for administration in the participant’s native language in compliance with IRB standards (IRB#20220835), and then back translated to ensure accurate translation by a certified translator” (page 7, lines 129-132).

3. (Page 6-7, line 120 -162) Authors should reorganize the manuscript under “Materials and methods” using level two subheadings for clarity and organization. Specifically, the following subheadings should be included:

1. Research Approach and Design: (Clearly define whether the study employed a quantitative approach using a cross-sectional design.)

Thank you. The Materials and methods section was broken down by the 4 suggested subheadings and extensively edited. We more clearly specified that “This was a cross-sectional study utilizing a quantitative approach” (page 7, lines 124-125).

2. Population, Sample Size, and Sampling: (Define the target population, explain how the sample size was determined using a specific formula with clear parameters, and state the sampling method used to select participants, justifying the choice of sampling method.)

Thank you for this suggestion. We updated information about sample size calculations and the sampling method as follows: “Statistical power was assessed using Pass2020 with an alpha rate of .05. We examined power for sample sizes from 150 to 200, using Cohen’s w as the effect-size measure with a sample of 173. There is 80% power to uncover a chi-square with 1 degree of freedom with a w of .213. Cohen (1988) characterized a w=.10 as a small and a w=.30 as a medium effect, so this study has 80% power to uncover a small-to medium effect” (pages 7-8, lines 145-149).6

3. Measures: (Describe and outline the items adapted from previously validated scales, report the internal consistency (Cronbach’s alpha coefficient) demonstrated by previous studies adopted the same scales, and include the internal consistency established by the current study.)

Thank you for the suggestion. Questions were adapted from previously validated scales: “The HIV transmission knowledge questions were adapted from the Marcelin et al. and Herek et al. studies7,8 and the personal/perceived stigma questions were adapted from the STRIVE stigma questionnaire.9 The Herek et al. study, where the majority of transmission knowledge survey questions were derived, had a high internal consistency (α = .77-.79)” (page 8, lines 153-156). We calculated the internal consistency for our study by separately grouping the transmission knowledge, treatment knowledge, and stigma questions. This revealed Cronbach’s alpha coefficients of 0.47, 0.75. and 0.36 for transmission knowledge, treatment knowledge, and stigma respectively. We believe these numbers are low given that these question subgroups only contain 3-6 questions, which was intentional to prevent respondent fatigue. The goal of our research was to improve our understanding of patient education/ attitudes on numerous topics, and thus we would not anticipate a high internal consistency. We have focused on item-level analysis (and adequacy of knowledge) as now described in the methods (page 8, lines 156-157).

4. Analysis: (Detail the statistical methods used to analyze the data, including any software utilized, the specific tests conducted, and the criteria for significance.)

Thank you. This section more thoroughly describes the software utilized and tests performed: “Qualtrics was used to collect participant responses, and then the raw data was exported to Microsoft Excel. Response rates to each survey question were then aggregated in Excel according to the methods above and imported into GraphPad prism for statistical analysis. All data underwent chi-square analysis using 1 degree of freedom at an alpha level of 0.05” (page 9, lines 182

---

## [Decision Letter · Decision Letter 1]

5 Jun 2025

Dear Dr. Sternberg,

Thank you for submitting your manuscript to PLOS ONE. After careful consideration, we feel that it has merit but does not fully meet PLOS ONE’s publication criteria as it currently stands. Therefore, we invite you to submit a revised version of the manuscript that addresses the points raised during the review process.

We look forward to receiving your revised manuscript.

Kind regards,

Zypher Jude G. Regencia, Ph.D.

Academic Editor

PLOS ONE

Reviewers' comments:

Reviewer's Responses to Questions

**Comments to the Author**

Reviewer #1: (No Response)

Reviewer #2: All comments have been addressed

2. Is the manuscript technically sound, and do the data support the conclusions?

Reviewer #1: Partly

Reviewer #2: Yes

3. Has the statistical analysis been performed appropriately and rigorously?

Reviewer #1: No

Reviewer #2: Yes

4. Have the authors made all data underlying the findings in their manuscript fully available?

Reviewer #1: Yes

Reviewer #2: Yes

5. Is the manuscript presented in an intelligible fashion and written in standard English?

Reviewer #1: No

Reviewer #2: Yes

Reviewer #1: Comments to the Authors:

Thanks for addressing majority of the previous-round’s review comments. This second-round feedback focuses on issues not previously addressed.

Additional major concerns:

Line 44–49, Page 3:

- “Transmission knowledge was significantly influenced by race” and “Familiarity with PrEP and/or PEP was also influenced by race…”.

- The use of “influenced” implies a causal relationship, which is not supported by a cross-sectional design and Chi-square test employed.

- Gold standard way of reporting, which reflects what Chi-square actually tests “Familiarity with PrEP and/or PEP was significantly associated with race…”

Line 217, Page 12 and throughout results section:

- Some chi-square statistics use more than two decimal places including for p-values.

- I strongly suggest consistency in rounding to two for chi-square statistic and three decimals for p-values throughout the manuscript.

Line 151-179, Page 8-9, Measures:

- The measures section should use consistent tense and structured subheadings

- Present each construct separately and clearly with the name and description of the instrument employed to measure the construct. For example:

HIV Transmission Knowledge and HIV stigma are the main constructs measured using previously validated instruments.

- Describe the origin of items, psychometric properties, scoring, and interpretation.

- Clearly state how variables were operationalized and analysed.

- There is no mention of Cronbach’s alpha coefficient for the adapted versions in current study. Given that the instruments were translated and adapted, I recommend that the authors either include a table summarizing the internal consistency (e.g., Cronbach’s alpha) for each scale or report these values directly under the Measures section to ensure transparency on the reliability of the instruments.

Line 167–169, Page 8-9:

- Defining adequate HIV transmission knowledge as missing ≤1 question is an arbitrary without statistical rationale.

- Provide justification from prior literature or explain how pilot data support this threshold as valid (acknowledge this as a limitation).

Line 237–240, Page 13:

- Authors claim that “there was no significant difference in rates of college educated individuals broken down by race or language” yet no statistical test or supporting data.

- Provide chi-square results that support this claim

- The authors mention race and language when exploring potential confounders, yet none of the central variables (HIV knowledge, stigma, PrEP/PEP awareness) were included in this analysis, making the confounder exploration to be methodologically unsound.

Figure 3 and Lines 231–233, Page 13:

- Some subgroup analyses, such as bilingual or Creole speakers, include very small sample sizes (n=9–11).

- Concern is that these small groups may produce unstable estimates and wide margins of error, which may invalidate statistical inferences.

- Acknowledge this limitation explicitly in the Discussion and temper your conclusion based on these groups.

Line 251-252, Page 14:

- Authors mentioned that “Otherwise, perceived stigma was not significantly influenced by race, country of origin, or primary language.” Yet without chi-square statistics.

Absence of adjusted analysis:

- All findings rely solely on bivariate chi-square analyses.

- Bivariate associations do not account for confounding exploration mentioned in the manuscript.

- Consider conducting a multivariate logistic regression for key outcomes or justify the decision not to do so in the limitations.

Minor concerns:

- Recommendation for language editing.

- Line 183 -184, page 9 – “All data underwent chi-square analysis…” avoid using passive voice in stead of active voice. “All data were analysed using chi-square…”

- Use of “unfamiliarity” consider using “lack of familiarity”

Line 345 and 351, Page 19:

- Reference 4 (Duplicate of Ref. 1):

- “Statistics Overview | Statistics Center | HIV/AIDS | CDC. Accessed April 15, 2024.https://www.cdc.gov/hiv/statistics/overview/index.html”

Reviewer #2: The manuscript addresses a critical and timely public health issue—persistent misconceptions and stigma surrounding HIV transmission and prevention.The study sample includes a demographically diverse group, with analyses disaggregated by race, language, and country of origin. This allows for the identification of disparities in knowledge and stigma across subgroups, particularly highlighting gaps among Black and White Hispanic participants and Spanish-speaking individuals. The revised manuscript clearly incorporates detailed responses to reviewer comments, including a comprehensive power analysis, clear inclusion/exclusion criteria, and improved internal consistency reporting. The restructuring of the Methods section using standardized subheadings enhances clarity and reproducibility.

**Do you want your identity to be public for this peer review?** For information about this choice, including consent withdrawal, please see our Privacy Policy

Reviewer #1: No

Reviewer #2: **Yes: ** BIEN ELI NILLOS

---

## [Author Response · Author response to Decision Letter 2]

10 Jul 2025

Thank you for reviewing. Your comments have greatly improved the manuscript.

Reviewer 1

Line 44–49, Page 3:

- “Transmission knowledge was significantly influenced by race” and “Familiarity with PrEP and/or PEP was also influenced by race…”.

- The use of “influenced” implies a causal relationship, which is not supported by a cross-sectional design and Chi-square test employed.

- Gold standard way of reporting, which reflects what Chi-square actually tests “Familiarity with PrEP and/or PEP was significantly associated with race…”

Thank you for this suggestion. The wording has been corrected throughout the manuscript so that “influenced” was replaced with “associated with” to better describe the non-causal relationship (lines 45, 48, 329, 361, and 379).

Line 217, Page 12 and throughout results section:

- Some chi-square statistics use more than two decimal places including for p-values.

- I strongly suggest consistency in rounding to two for chi-square statistic and three decimals for p-values throughout the manuscript.

Thank you for this suggestion. Rounding is now consistent throughout the manuscript to two decimals for chi-square statistics and three decimals for p-values.

Line 151-179, Page 8-9, Measures:

- The measures section should use consistent tense and structured subheadings

- Present each construct separately and clearly with the name and description of the instrument employed to measure the construct. For example:

HIV Transmission Knowledge and HIV stigma are the main constructs measured using previously validated instruments.

- Describe the origin of items, psychometric properties, scoring, and interpretation.

- Clearly state how variables were operationalized and analysed.

- There is no mention of Cronbach’s alpha coefficient for the adapted versions in current study. Given that the instruments were translated and adapted, I recommend that the authors either include a table summarizing the internal consistency (e.g., Cronbach’s alpha) for each scale or report these values directly under the Measures section to ensure transparency on the reliability of the instruments.

Thank you for your comments. Subsection headings were added, and we updated verb tense throughout the section. We updated information to better describe the origin of questionnaire items: “HIV transmission knowledge and HIV stigma are the main constructs measured using previously validated instruments. One third of the HIV transmission knowledge questions are adapted from the Marcelin et al. study and two thirds of the transmission knowledge questions are derived from the Herek et al. study. All of the personal/perceived stigma questions are adapted from the STRIVE stigma questionnaire. The Herek et al. study, has a high internal consistency (α = .77-.79) but there are no reported internal consistencies reported for the Marcelin et al. or STRIVE stigma questionnaires” (page 8, lines 188-194).

A new subsection was incorporated to better define survey scoring; “Transmission knowledge scores are classified as adequate if participants missed no more than one question out of the six total. This cut off was determined based on pilot data from the initial health fair survey responses (n=12), which demonstrated a mean of 80% correct. Participants are considered familiar with PrEP/PEP if they noted familiarity with either medication on the questionnaire. For the four stigma questions, participants demonstrated personal stigma if they screened positive on one of the first two questions or perceived stigma if they screened positive on at least one of the last two questions” (page 9, lines 274-280).

Thank you for your recommendation to report the Cronbach’s alpha coefficient for our adapted survey. We included this information in the measures section to ensure transparency of the reliability of the survey: “We calculated the internal consistency of our study by performing Cronbach’s alpha analysis on PEP/PrEP treatment knowledge questions and calculated alpha coefficients of 0.75.” (page 10, lines 308-311). We did not report an alpha for HIV knowledge because our survey included true-false statements and the number correct is a clear indicator of a person’s knowledge of the topic. The 2-item stigma scales had a low alpha value and thus did not make a reliable scale. Therefore, we removed the chi-square findings and acknowledged this limitation in the results: “Perceived and personal stigma had a low Cronbach’s alpha coefficient value and thus we drew limited conclusions about their association with race, country of origin, and primary language” (page 15, lines 261-262)

Line 167–169, Page 8-9:

- Defining adequate HIV transmission knowledge as missing ≤1 question is an arbitrary without statistical rationale.

- Provide justification from prior literature or explain how pilot data support this threshold as valid (acknowledge this as a limitation).

Thank you for this suggestion. The rationale for determining the cut-off was determined from pilot data: “Transmission knowledge scores are classified as adequate if participants missed no more than one question out of the six total. This cut off was determined based on pilot data from the initial health fair survey responses (n=12), which demonstrated a mean of 80% correct” (page 9, lines 274-276). Information about this limitation was added to the discussion section: “When evaluating survey responses, we utilized pilot data to determine the cut-off for sufficient transmission knowledge among the population which introduces variability” (page 18, lines 490-492).

Line 237–240, Page 13:

- Authors claim that “there was no significant difference in rates of college educated individuals broken down by race or language” yet no statistical test or supporting data.

- Provide chi-square results that support this claim

- The authors mention race and language when exploring potential confounders, yet none of the central variables (HIV knowledge, stigma, PrEP/PEP awareness) were included in this analysis, making the confounder exploration to be methodologically unsound.

Thank you for these comments. The manuscript was updated to include the non-significant chi-square statistics: “There was no significant difference in the rates of college educated individuals broken down by race (χ²(2, N = 116)= 3.85, p =.146), language (χ²(1, N = 97)= 0.01, p =.938), or country of origin (χ²(1, N = 98)= 0.68, p =.408)” (page 14, lines 371-373). We removed the statement about exploring potential confounders.

Figure 3 and Lines 231–233, Page 13:

- Some subgroup analyses, such as bilingual or Creole speakers, include very small sample sizes (n=9–11).

- Concern is that these small groups may produce unstable estimates and wide margins of error, which may invalidate statistical inferences.

- Acknowledge this limitation explicitly in the Discussion and temper your conclusion based on these groups.

Thank you for this suggestion. We did not draw conclusions about bilingual or Creole speakers given the small sample sizes. This limitation was added to the discussion section: “Moreover, we could not adequately assess bilingual or Haitian-Creole speakers given the low sample size” (page 18, lines 485-486).

Line 251-252, Page 14:

- Authors mentioned that “Otherwise, perceived stigma was not significantly influenced by race, country of origin, or primary language.” Yet without chi-square statistics.

Absence of adjusted analysis:

- All findings rely solely on bivariate chi-square analyses.

- Bivariate associations do not account for confounding exploration mentioned in the manuscript.

- Consider conducting a multivariate logistic regression for key outcomes or justify the decision not to do so in the limitations.

Thank you for the suggestions. The chi-square statistics are now reported in the manuscript: “Otherwise, perceived stigma was not significantly associated with race (χ²(2, N = 163)= .56, p =.757), country of origin (χ²(1, N = 137)= 1.76, p =.185), or primary language (χ²(1, N = 138)= .62, p =.432)” ( page 15, line 399-401). We justified our reasons for running chi-square analysis rather than a multivariable logistic regression in the discussion section: “While we assessed interactions via bi-variate chi-square analysis, we did not run a multivariable logistic regression to assess confounders because numerous demographic variables including race, country of origin, and language are interrelated and cannot be considered independent variables” (page 18, lines 493-496).

Minor concerns:

- Recommendation for language editing.

- Line 183 -184, page 9 – “All data underwent chi-square analysis…” avoid using passive voice instead of active voice. “All data were analysed using chi-square…”

- Use of “unfamiliarity” consider using “lack of familiarity”

Thank you. This has been corrected. (lines 54, 350, 367).

Line 345 and 351, Page 19:

- Reference 4 (Duplicate of Ref. 1):

- “Statistics Overview | Statistics Center | HIV/AIDS | CDC. Accessed April 15, 2024.https://www.cdc.gov/hiv/statistics/overview/index.html”

Thank you. The duplicate citation has been deleted from the manuscript.

---

## [Editor Report · Decision Letter 2]

21 Jul 2025

HIV Knowledge and Reported Stigma Among South Florida Health Fair Participants

PONE-D-24-36376R2

Dear Dr. Sternberg,

We’re pleased to inform you that your manuscript has been judged scientifically suitable for publication and will be formally accepted for publication once it meets all outstanding technical requirements.

Kind regards,

Zypher Jude G. Regencia, Ph.D.

Academic Editor

PLOS ONE
---

## [Editor Report · Acceptance letter]

PONE-D-24-36376R2

PLOS ONE

Dear Dr. Sternberg,

I'm pleased to inform you that your manuscript has been deemed suitable for publication in PLOS ONE. Congratulations! Your manuscript is now being handed over to our production team.

Kind regards,

on behalf of

Dr. Zypher Jude G. Regencia

Academic Editor

PLOS ONE